# Single-cell fluidic force microscopy reveals stress-dependent molecular interactions in yeast mating

Marion Mathelié-Guinlet 1, Felipe Viela 1, Jérôme Dehullu1, Sviatlana Filimonava2, Jason M. Rauceo2, Peter N. Lipke 3✉ & Yves F. Dufrêne 1✉

Sexual agglutinins of the budding yeast *Saccharomyces cerevisiae* are proteins mediating cell aggregation during mating. Complementary agglutinins expressed by cells of opposite mating types "**a**" and "α" bind together to promote agglutination and facilitate fusion of haploid cells. By means of an innovative single-cell manipulation assay combining fluidic force microscopy with force spectroscopy, we unravel the strength of single specific bonds between **a**- and α-agglutinins (~100 pN) which require pheromone induction. Prolonged cell–cell contact strongly increases adhesion between mating cells, likely resulting from an increased expression of agglutinins. In addition, we highlight the critical role of disulfide bonds of the **a**-agglutinin and of histidine residue $H_{273}$ of α-agglutinin. Most interestingly, we find that mechanical tension enhances the interaction strength, pointing to a model where physical stress induces conformational changes in the agglutinins, from a weak-binding folded state, to a strong-binding extended state. Our single-cell technology shows promises for understanding and controlling the complex mechanism of yeast sexuality.

[1] Louvain Institute of Biomolecular Science and Technology, UCLouvain, Croix du Sud, 4-5, bte L7.07.07, 1348 Louvain-la-Neuve, Belgium. [2] Department of Sciences, John Jay College of the City University of New York, New York, NY 10019, USA. [3] Biology Department, Brooklyn College of the City University of New York, 2900 Bedford Avenue, Brooklyn, NY 11210, USA. ✉email: PLipke@brooklyn.cuny.edu; yves.dufrene@uclouvain.be

Like prokaryotic cells, yeasts rely on cell surface adhesion proteins (adhesins) to selectively interact with their environment[1]. Yeast adhesins participate in biofilm formation and attachment to host cells, being therefore critical for pathogenesis. Interestingly, they have also evolved to facilitate fungal development. A prototypical example of yeast adhesins are sexual agglutinins that mediate cell–cell contacts, the first initial step in the cell mating process that leads to fusion of haploid cells of opposite mating type into diploid cells[2].

Sexual agglutination of the bread yeast *Saccharomyces cerevisiae* involves the heterophilic interaction between two cell surface glycoproteins, the agglutinins "α-Ag" and "**a**-Ag", expressed by haploid *MAT*α and *MAT***a** cells, respectively[2–4]. The α-agglutinin is a single polypeptide of 650 amino acids including 19 residues for a signal peptide and a signal for glycosyl-phosphatidylinositol (GPI) anchor addition. α-Agglutinin structure is highly reminiscent of the Als adhesin family of *Candida albicans*[5]. Indeed, it consists of a highly glycosylated stalk C-terminal (~300 amino acids) that anchors the protein to the wall and a β-sheet-rich N-terminal immunoglobulin (Ig)-like region homologous to Als N-terminal domains. This N-terminal region contains the binding site for **a**-agglutinin[6,7]. The **a**-agglutinin is a dual subunit protein with Aga1p (725 residues) anchoring the receptor-binding glycopeptide Aga2p (69 residues) onto the cell surface, through two disulfide bonds[8–11]. Both agglutinins are covalently anchored to cell wall polysaccharide by cleaving the GPI glycan and cross-linking the remnant to cell wall polysaccharide, a process analogous to sortase anchoring of adhesins in Gram-positive bacteria[1,6].

Each mating type also produces a peptide sex pheromone, which induces mating behavior in the opposite mating type. *MAT*α cells produce the pheromone α-factor, and *MAT***a** cells produce a pheromone called **a**-factor. Haploid cells also express a receptor specific for the pheromone from the other mating type. The receptors are G protein-coupled receptors, linked to a protein kinase cascade that leads to altered gene expression. As a consequence, each mating type responds to the pheromone from the other mating type, increasing expression of pheromone, receptor, and agglutinin specific to that mating type. The agglutinins are constitutively expressed at low levels, so pheromone production and response are critical for increasing surface agglutinin, especially in *MAT***a** cells. Thus pheromone treatment leads to robust cellular aggregation when cells of both mating types are mixed, leading to high frequency mating[12].

Agglutinins have been extensively investigated by biochemical and genetic approaches, especially those involved in *S. cerevisiae* mating.[1,2] On one hand, the histidyl residue His$_{273}$ of α-Ag has been shown to be responsible for the binding of *MAT***a** to *MAT*α cells[11], yet some other regions of α-Ag, spatially close to this residue, have also been identified as potentially involved in such interactions[6]. On the other hand, the 10 C-terminal amino acids of Aga2p are also required for agglutination with α cells[10]. These **a**–α interactions are actually highly complex, involving several regions of Aga2p and α-Ag that bind with both nanomolar and micromolar affinities[9,13]. Indeed, it has been suggested that the initial interaction of **a**- and α-agglutinins triggers conformational changes in both agglutinins resulting in a tight and irreversible binding[13]. Specifically, the anchorage subunit Aga1p is critical for maintaining Aga2p in its active conformation for the interaction with α-agglutinin[9]. The C-terminal peptide of Aga2p inserts in the binding cleft of α-Ag, in the same way as the tight binding site for C-terminal peptide ligands in Als proteins, in, e.g., Als3p[14]. Such interaction typically illustrates a β-strand complementation where the C-terminal part of Aga2p becomes part of a β-sheet in the α-Ag Ig domains.

Despite these structural data gained on the determinants of the *MAT*α and *MAT***a** cell agglutination, the molecular mechanisms

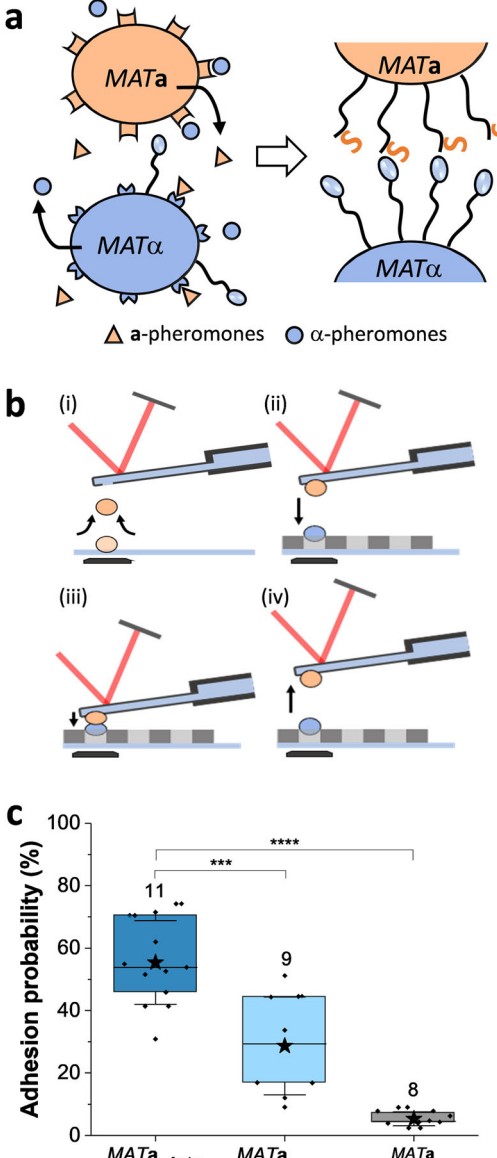

**Fig. 1 FluidFM as a single-cell manipulation tool to study sexual agglutination. a** Scheme of the mating processes between *Saccharomyces cerevisiae MAT***a** and *MAT*α cells. Mating occurs through the specific interaction of **a**- and α-agglutinins whose surface expression is enhanced when *MAT***a** are exposed to α-pheromones and *MAT*α are exposed to **a** pheromones. **b** Scheme of the fluidFM set-up showing the main steps for single cell–cell measurements. (i) A single *MAT***a** cell is first immobilized at the aperture of the fluidFM cantilever by applying a negative pressure. (ii) The single-cell probe is then brought into contact with a single *MAT*α cell trapped in a porous membrane, (iii) being either or not maintained in contact for 30 min, and (iv) finally retracted to quantify cell–cell adhesion forces. **c** Box plots of the adhesion probability between *MAT***a** and *MAT*α cells showing the key role of pheromones treatment: (i) both mating types were treated with the opposite mating pheromones (*MAT***a**$_{α-factor}$ and *MAT*α$_{a-factor}$), (ii) only *MAT*α cells were treated with *MAT***a** pheromones (*MAT***a** and *MAT*α$_{a-factor}$) and (iii) none of the cells were pretreated with pheromones (*MAT***a** and *MAT*α). Stars are the mean values, lines the medians, boxes the 25–75% quartiles, and whiskers the SD from *N* independent cell pairs (*N* is indicated above the corresponding box). Student's *t* test: ***$p ≤ 0.001$, ****$p ≤ 0.0001$.

driving the interactions between α-Ag and **a**-Ag remain elusive. The development of front-end tools, which can probe individual molecules on live cells, enable getting insights into the strength and dynamics of such processes otherwise inaccessible by traditional ensemble bio-assays[15]. Among these, fluidic force microscopy (FluidFM) combines high-throughput nanofluidic manipulation of single cells with the high force sensitivity of atomic force microscopy (AFM)[16–21]. Here we use an original approach combining FluidFM with force clamping, which unlike traditional AFM assays, enable us to hold cell pairs in contact in a controlled way and to generate larger data sets. This method allows us to unravel the molecular forces involved in yeast agglutination, highlighting the previously undescribed role of mechanical stress in controlling the strength of sexual agglutination in yeasts.

## Results

**FluidFM as a tool to study sexual agglutination.** In single-cell force spectroscopy, cells are generally immobilized on AFM cantilevers using (bio)chemical treatments, which are invasive and/or poorly controlled. FluidFM enabled us to quickly and reversibly immobilize single live cells on the cantilever tip and probe the forces toward a target cell (Fig. 1). As a proof of concept, we tested the ability of the method to show that sex pheromones induce yeast agglutination, on a single-cell basis. We investigated the agglutination forces between *MAT*a and *MAT*α cells, either exposed or not to the sex pheromones of their mating partner (Fig. 1a). Cells were grown separately overnight; then the growth medium of *MAT*a cells, containing the secreted pheromone **a**-factor, was switched or not with the growth medium from *MAT*α cells containing the sex pheromone α-factor, and both cells were re-incubated at 30 °C for 30 min. A single *MAT*a

cell, physically immobilized at the apex of the microchanneled cantilever, was brought into contact with an individual *MAT*α cell, mechanically trapped in a porous membrane (Fig. 1b), and cell–cell adhesion was then quantified by force spectroscopy (Fig. 1c and see Supplementary Fig. 1 for adhesion distributions of representative cells). While the adhesion probability, i.e., the frequency of adhesion events recorded in a 100-by-100 nm$^2$ area on the probed cell, was extremely low when cells were not exposed to pheromones (5 ± 2%, mean ± SD, 8 independent cell pairs), it increased to ~50% (55 ± 13%, mean ± SD from 11 independent cell pairs) for pheromone-treated cells, confirming the crucial role of pheromones in the specific binding between **a**- and α-agglutinins. Moreover, when only one mating type, *MAT*α, was incubated with the pheromones of the mating partner, *MAT*a, the cell–cell adhesion remains lower (28 ± 16%, mean ± SD from 9 independent cell pairs) suggesting that both cells require their partner pheromones to express, at a sufficient level, their constitutive agglutinins that will lead to their further co-adhesion. This frequency is less than expected from the converse experiment (untreated *MAT*α cells with pheromone-treated *MAT*a cells), because pheromone treatment increases surface levels of α-agglutinins about 2-fold, whereas pheromone treatment of *MAT*a cells increases **a**-agglutinin expression 10–20-fold[2].

**The binding strength of single agglutinins is around 100 pN.** Using this approach, we sought to investigate the binding forces between individual **a**- and α-agglutinins with cells that had been treated with the relevant pheromones (Fig. 2). The time of co-incubation was optimized to 30 min, which led to the highest probability of adhesion between both mating yeasts. Unbinding force distributions featured a broad spectrum from 50 to 800 pN

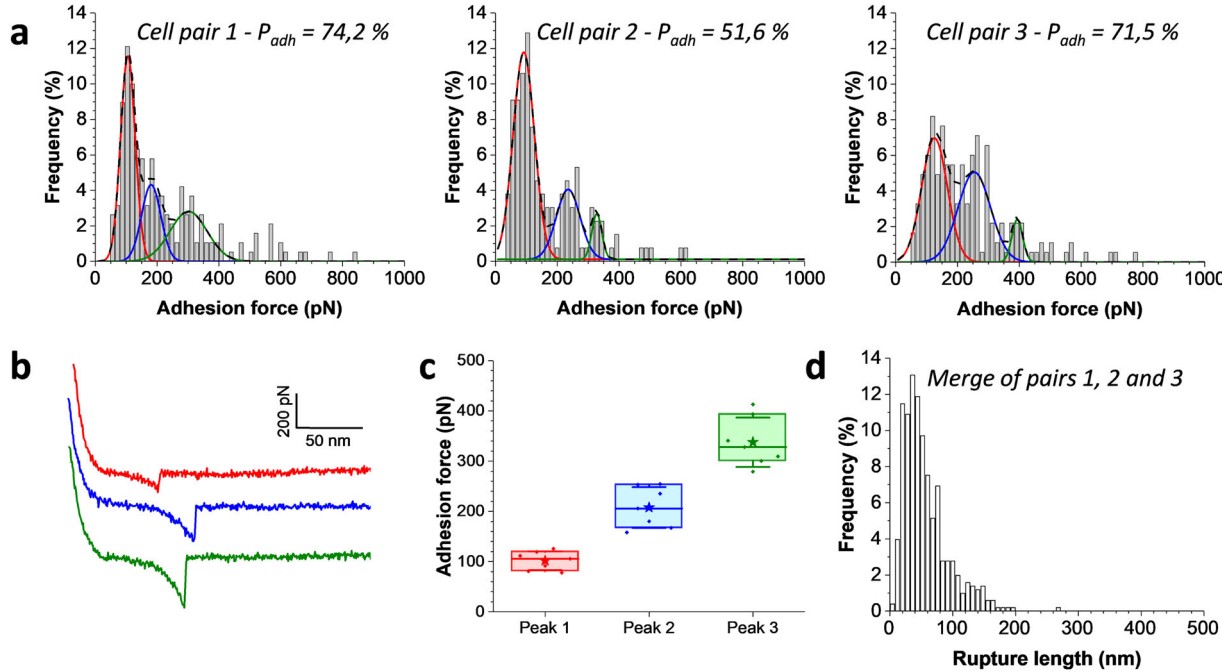

**Fig. 2 The binding strength of single agglutinins. a** Maximum adhesion force histograms obtained by recording force–distance curves in PBS between three independent *MAT*a and *MAT*α cell pairs pretreated with pheromones, at an applied force of 1 nN, with a probing time of 1 s and a retraction velocity of 1000 nm s$^{-1}$. The corresponding adhesion probability ($P_{adh}$) is mentioned in each histogram. Gaussian fits for a multimodal distribution are overlapped with the raw histogram: in colored plain lines, the independent Gaussians and in dashed black lines the overall fit. **b** Representative retraction force profiles color coded according to the peaks identified in the distributions. **c** Box plots of the mean adhesion forces revealed by the multi-peak center analysis, as illustrated in **a**. Stars are the mean values, lines the medians, boxes the 25–75% quartiles, and whiskers the SD from $N = 7$ independent cell pairs. **d** Rupture length histogram obtained by merging the data obtained on the three cell pairs shown in **a**.

(Fig. 2a) with most of the curves exhibiting single adhesion force peaks (Fig. 2b). The majority of cell pairs showed a force distribution that could be fitted with Gaussians (Fig. 2a) centered at $102 \pm 19$, $208 \pm 41$, and $338 \pm 49$ pN, respectively (Fig. 2c). This suggests that the 100-pN unit force might correspond to the strength of a single bond, while larger forces would correspond to multiple interactions. Variations were observed from one cell pair to another likely resulting from differences in agglutinin expression.

For all interactions, rupture lengths were rather short, i.e. $58 \pm 35$ nm (mean ± SD, $n = 432$ adhesive events from 3 independent cell pairs; Fig. 2d). Previous studies have revealed that only the Aga2p (69 amino acids) and the Ig-like domains of the N-terminal of α-agglutinin (~300 amino acids) are physically involved in the interaction. Given that each amino acid contributes to 0.36 nm of the contour length of an extended protein[22], a fully extended and unfolded molecular complex should give rise to ~130 nm extensions. However, because Aga2p is largely unstructured, only the last 10 amino acids in Aga2p are expected to be extended by force (~3.6 nm). In contrast, the binding domain of α-agglutinin is highly folded and stabilized by disulfide bonds[7], and this restraint limits extension to about 50 nm. Thus the amount of stretch is close to what we expect from stretching the cell–cell bond.

To confirm that the ~100 pN force is due to the rupture of single bonds, we lowered the applied force and the probing time, i.e., the time spent in contact with the probed cell while doing the approach–retract cycles (Fig. 3)[23]. The two parameters influenced significantly the adhesion probability between cells (Fig. 3a), which dropped from $64 \pm 19$ to $32 \pm 19\%$ when reducing the applied force to 0.25 nN and to $26 \pm 5\%$ when reducing the probing time to 0.1 s (mean ± SD from 5 independent cell pairs). Decreasing both the applied force and probing time led to extremely low adhesion probability and to statistically unreliable force distributions, and so the resulting data were not further interpreted. Rupture force distributions obtained under various probe times and force regimes again showed a relatively broad spectrum (see Supplementary Fig. 2 for adhesion distributions of one representative cell). Adhesion forces were categorized by quartiles, namely, the forces at which 25% (Q1), 50% (median), and 75% (Q3) of the adhesion events are observed (Fig. 3b). Decreasing the probing time, and even more strikingly the applied force, led to a decrease in the values of Q1, median, and Q3, while the minimum observed forces remained the same. So there was a shift toward the population of low forces when the applied force or duration was decreased. Moreover, the proportion of the first peak ($102 \pm 19$ pN) in the force distributions slightly increased when lowering the applied force and remained constant when reducing the probing time (Fig. 3c), while the second and third peaks were not impacted. This supports the idea that reducing the applied force, thus the cell–cell contact area, favors single interactions between a- and α-agglutinins.

**Prolonged contact between mating cells increase agglutination forces.** Cell–cell contact is crucial in facilitating mating and fusion of yeast partners. We therefore wondered whether agglutinin interactions are modulated by increasing the duration of cell–cell contact, i.e., the time that cells were "artificially" forced to be in contact independently of the probing time, that is, the time required to record a force–distance curve (defined above). Force clamp was used to maintain *MAT*a and *MAT*α cells in close proximity for 30 min with a controlled and constant applied force of 1 nN. Such extended contact did not impact the shape of adhesion profiles, a broad distribution of forces from 50 to 800 pN being still observed (Fig. 4a, b). However, the adhesion

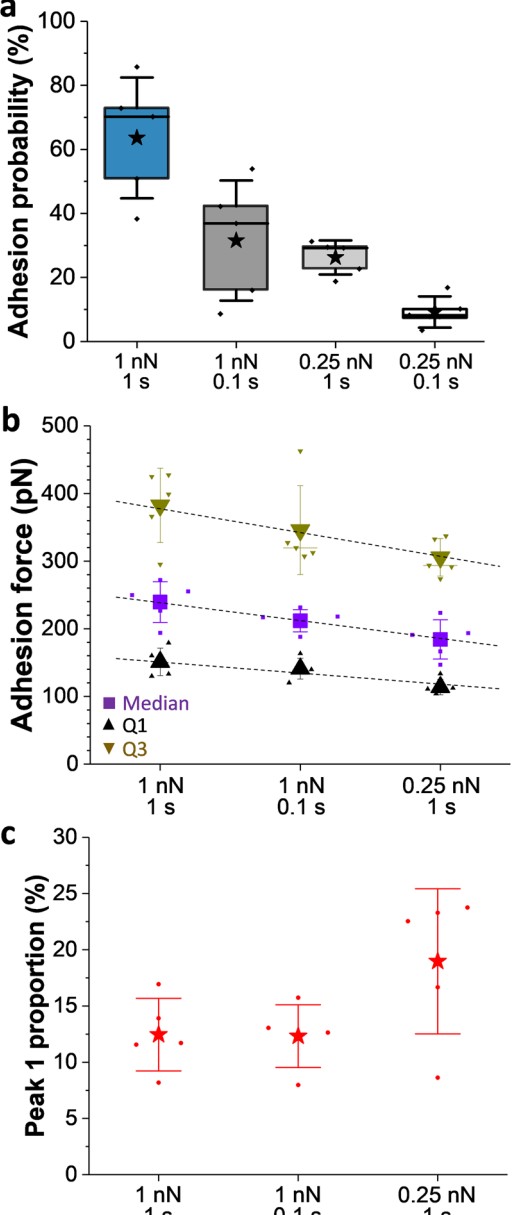

Fig. 3 The role of applied force and probing time in cell–cell adhesion. a Box plots of the adhesion probability between *MAT*a and *MAT*α cells, pretreated with pheromones, recorded under different applied forces (1 or 0.25 nN) and with different probing times during force–distance curve measurements (1 or 0.1 s). Stars are the mean values, lines the medians, boxes the 25-75% quartiles, and whiskers the SD from $N = 5$ independent cell pairs. b Scatter plots, overlapped with data points, of the adhesion forces reached at diverse quartiles (Q1: 25%, median: 50% and Q3: 75% of the adhesive population) as a function of the applied force and probing time used to record the force–distance curves between *MAT*a and *MAT*α cells. Error bars are the SD from $N = 6$ independent cell pairs. Dashed lines are a guide for the eye. c Proportion of the first peak identified in Fig. 2c (vs the total adhesive population) as a function of the same parameters. Error bars are the SD from $N = 5$ independent cell pairs.

probability increased from $54 \pm 13$ to $75 \pm 13\%$ (mean ± SD from 10 independent cell pairs, Fig. 4c), highly suggesting that initial contact between *MAT*a and *MAT*α cells enhances surface expression of agglutinins and their interaction required for efficient mating. This prompted us to reconsider closely the force distributions obtained before and after the 30-min contact

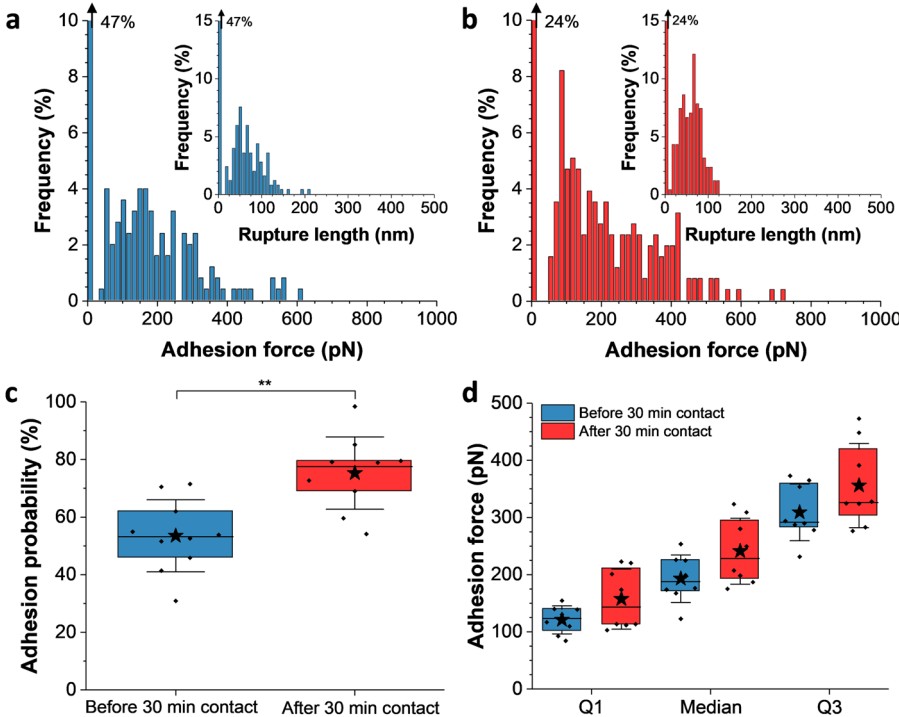

**Fig. 4 Extended cell–cell contact increases agglutination forces.** Maximum adhesion force histograms **a** before and **b** after a prolonged contact of 30 min between cells, obtained by recording force–distance curves in PBS between a representative *MAT***a**–*MAT*α cell pair pretreated with pheromones, at an applied force of 1 nN, with a probing time of 1 s and a retraction velocity of 1000 nm s⁻¹. Insets show the corresponding distribution of rupture lengths. Percentage on the top left corner stands for the non-adhesive events. **c** Box plots of the adhesion probability between *MAT***a** and *MAT*α cells before and after 30 min contact ($N = 10$ independent cell pairs). **d** Box plots of the adhesion forces reached at diverse quartiles (Q1: 25%, median: 50% and Q3: 75% of the adhesive population) before and after 30 min contact ($N = 8$ independent cell pairs), suggesting a slight shift toward higher force populations. Stars are the mean values, lines the medians, boxes the 25–75% quartiles, and whiskers the SD from $N = 8$ independent cell pairs. Student's *t* test: **$p \leq 0.01$.

between cells, in light of the parameters mentioned above (Fig. 4d). Averaging results obtained on 8 independent cell pairs after prolonged contact showed that all quartiles describing the force distributions (Q1, median, and Q3) tended to slightly increase (for independent cell pairs, see Supplementary Fig. 3), implying that high forces get more populated as compared to low forces (~100 pN). This led us to believe that increasing the duration of cell–cell contact favors surface expression of agglutinins on opposite mating partners and might favor multivalent interactions that, in turn, would help mating and subsequent fusion of diploid cells. Of note, rupture lengths were not significantly modified after prolonged contact (66 ± 11 nm vs 64 ± 11 nm, mean ± SD from 10 independent cell pairs), emphasizing that, if multiple bonds break, they do so in parallel rather than sequentially (see, e.g., Fig. 4a, b insets).

**Molecular origin of a- and α-agglutinin interaction.** To gain further insight into the molecular nature of the measured cell–cell adhesion forces, we injected specific agents that act on specific regions of the agglutinins (Fig. 5). As a reducing agent, dithiothreitol (DTT) acts on the disulfide bonds between the **a**-agglutinin subunits. A significant decrease in adhesion probability of about 10-fold was observed after injection of DTT at a final concentration of 10 mM (Fig. 5a, b). This confirms the hypothesis according to which disulfide bonds between the anchoring Aga1p subunit and the binding subunit Aga2p are required for cell–cell adhesion[9] (Fig. 5c) and demonstrates the essentiality of the Aga2p subunit, which is released from the cell surface by DTT treatment. This is in line with circular dichroism experiments revealing a decrease in β-strand content accompanied by an increase in α-helices upon DTT treatment leading to **a**-cells becoming non-

adhesive[13]. Diethyl pyrocarbonate (DEPC) treatment was also performed and led to a 10-fold drop in binding probability between *MAT***a** and *MAT*α cells (Fig. 5a, b), highlighting the critical role of the histidine H₂₇₃ of α-agglutinin in the binding to Aga2p subunit of its mating receptor (Fig. 5c).

**The interaction between a- and α-agglutinins is enhanced by mechanical stress.** We investigated the dynamics of the agglutinins interactions by measuring the unbinding force as a function of the loading rate (LR, rate at which the mechanical force is applied), varying the cantilever retraction velocity from 0.2 to 10 μm s⁻¹ (Fig. 6). LR values were extracted from the linear slope preceding the rupture event on the force vs time curves. The dynamic force spectrum showed a non-linear increase of unbinding force when increasing the LR over a wide range from 10² to 10⁶ pN s⁻¹ (Fig. 6a and see Supplementary Fig. 4 for distributions of one representative cell). Qualitatively this is in line with the notion that the rupture force between receptors and ligands increases with the rate at which force is applied. While the Bell–Evans theory considers a log-linear relationship between the LR and rupture force[24], the Friddle–Noy–de Yoreo model describes nonlinear trends in rupture forces, due to the reforming of single bonds at low LRs, in the close-to-equilibrium regime[25]. None of these models fitted our data, as extremely strong interactions (up to 1500 pN) were observed in higher proportions under high tensile loading, LRs >10⁵ pN s⁻¹ (Fig. 6b), suggesting that the strength of single bonds is enhanced under stress as for catch bonds. Note that multivalent interactions may also account for some of the high forces at high LRs, but this phenomenon clearly does not explain on its own the force-enhanced adhesion observed here. Lastly, we note that rupture lengths and adhesion

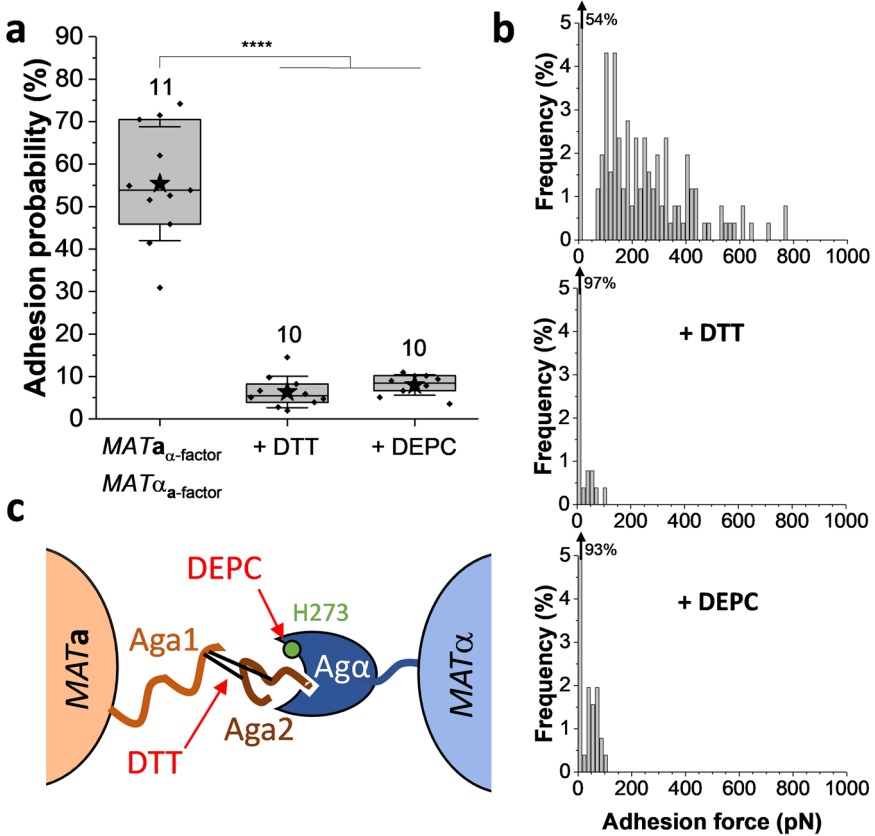

**Fig. 5 Agglutination forces critically depend on disulfide bonds and His$_{273}$ residues. a** Influence of DTT and DEPC treatments on the adhesion probability between $N$ independent $MAT\mathbf{a}$ and $MAT\alpha$ cell pairs ($N$ is indicated above the corresponding boxes) pretreated with pheromones. Stars are the mean values, lines the medians, boxes the 25–75% quartiles, and whiskers the SD from $N$ independent cell pairs. Student's $t$ test: ****$p \leq 0.0001$. **b** Representative unbinding force histograms before and after injection of DTT or DEPC. Percentage on the top left corner stands for the non-adhesive events. **c** Scheme of the molecular mechanism involved in the interaction between **a**- and α-agglutinins, showing the binding sites of both agglutinins and the regions targeted by DTT and DEPC.

forces both increased under force, suggesting that at high force the proteins likely become more extended.

## Discussion

The specific recognition between sexual cell adhesion proteins **a**- and α-agglutinins plays a central role during mating of *S. cerevisiae*. Despite decades of structural and biochemical studies on yeast agglutination, we still know little about the molecular details of the underlying interaction. Here we have quantified the strength and dynamics of single heterotypic interactions between **a**- and α-agglutinins, findings that were made possible with our innovative FluidFM force clamp assay allowing us to hold single-cell pairs in constant contact for extended times in a controlled way. Unlike in classical single-cell force spectroscopy, immobilization of individual cells onto the cantilever probe does not require any (bio)chemical treatment or drying and enables to generate larger numbers of data sets. Besides keeping the cells in contact, force clamping prevents any substantial drift and allows precise control of cell–cell localization and contact force for extended times (here 30 min), thus offering new prospects to study contact-dependent cell adhesion and cell signaling. The force-sensitive agglutination interaction unraveled here, never reported for a sexual adhesin, provides a means to yeast cells to strengthen their adhesive properties during mating and fusion. Our study shows that mechanobiology is important for yeast cell behavior, including ability to mate under mechanical stress.

The binding strength between single agglutinins is in the order of 100 pN. This corresponds to single specific bonds because: (i) they do not depend on the applied force or probing time and (ii) they are only observed when the cells are treated with the relevant sex pheromones. Increasing the duration of cell–cell contact to 30 min strongly influences the adhesion probability, resulting from an increase in the expression of agglutinins or from their structural reorganization at the cell surface. That the interaction requires pheromone induction, strengthens over time, and is abolished by treatments that inactivate the agglutinins all validate the idea that the reported molecular forces monitor the same interactions of α-agglutinin and **a**-agglutinin, that had been characterized biochemically earlier[9–11,13]. Single-molecule analyses also provide direct and quantitative pieces of evidence that disulfide bonds between the anchoring Aga1p subunit and Aga2p subunit of **a**-agglutinin, as well as His$_{273}$ of α-agglutinin, are required for agglutination.

Our main finding is that the agglutinin interaction strengthens under mechanical stress, with forces up to 1500 pN at LRs larger than $10^5$ pN s$^{-1}$. We speculate that this previously undescribed phenomenon involves force-induced conformational changes in either one or both agglutinins, from a weak-binding folded state to a strong-binding extended state. This model is reinforced by the increase in rupture length with unbinding force and by earlier work showing that conformational shifts in the agglutinins are essential for effective binding and that the agglutinins bind to each other through a specific and complex interaction consistent with at least two states, weak and tight[13]. Formation of tight

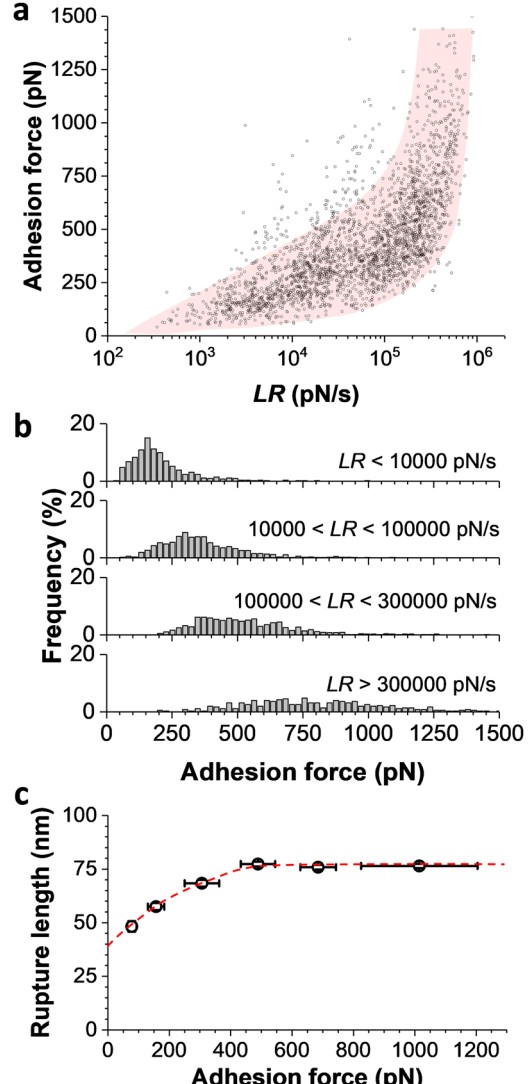

**Fig. 6 Physical stress strongly enhances agglutination forces between *MAT*a and *MAT*α cells. a** Dynamic force spectroscopic plot obtained by recording force–distance curves in PBS between *MAT*a and *MAT*α cells, with an applied force of 1 nN and a probing time of 1 s ($n = 2761$ data points from $N = 6$ independent cell pairs). **b** Corresponding unbinding force histograms as a function of discrete ranges of LRs, emphasizing the shift toward higher forces with increasing stress. **c** Scatter plot of the rupture lengths observed according to different ranges of unbinding forces. The dashed red line is a guide for the eye.

bonds is accompanied by a change in the secondary structure of the receptor α-agglutinin and also the ligand a-agglutinin. We suggest that initial interaction of the agglutinins triggers a conformational shift that greatly increases the contact area, and thus multipoint attachments, between the glycoproteins, resulting in tight binding. In addition, it is possible that, under force, cryptic high affinity sites become exposed to mediate strong cell–cell adhesion. This complexity adds to an earlier finding showing that mating-type-specific cell–cell recognition is a complex mechanism, involving several regions of Aga2p with several regions of Aga1p, showing here for the first time to our knowledge the role of force-induced conformational shifts in the complex formed by the two mating agglutinins.

To conclude, single-cell FluidFM allowed us to unravel the physical complexity of yeast–yeast interactions that was only approachable in a qualitative and anecdotal way before in standard bio-assays. That mating-type-specific agglutination is modulated in response to mechanical stress might be of biological and technological relevance. In natural and technological environments, the cells can experience a variety of shear forces, such as liquid flow. Force-sensitive agglutinin interactions might therefore be used by the cells to tune their adhesive properties during mating and fusion. Similar molecular behavior might find applications for cell engineering to ensure stability in synthetic biology experiments or substrate adhesion under flow in cellular industrial reactors[26–28].

## Methods

**Yeasts and culture.** *S. cerevisiae* BY4741 (*MAT*a) and BY4742 (*MAT*α) strains were separately cultivated on Yeast Peptone Dextrose (YPD) agar plates at 30 °C. One colony of each type was separately inoculated in liquid YPD medium. The cultures were incubated overnight at 30 °C under gentle agitation (180 rpm). Exposition to pheromones was performed by exchanging used growth medium between the two mating-type cells for 30 min.

**Preparation of FluidFM probes.** Cell probes were prepared using rectangular, hollow silicon nitride cantilever with an aperture of 4 μm and a nominal spring constant of 0.3 N m$^{-1}$ (Cytosurge AG). The micropipets were coated using a gas phase SigmaCote to limit protein adsorption and biofouling. Probes were placed in a dessicator connected to a vacuum pump. One-to-2 mL of the SigmaCote solution were added to the dessicator and the pressure was reduced to 20 kPa for 1 h. The probes were then oven dried at 80 °C for 1 h. Prior to measurement, FluidFM probes were calibrated using the thermal noise method. At the end of each experiment, contaminants and cell debris were washed from the cantilevers, by using solutions of Terg-a-zyme and NaOH and applying sequential negative and positive pressures. The cleaned probes were stored in pure water until the next experiment.

**Preparation of single-cell experiments.** Target cells (*MAT*α) were immobilized mechanically into porous membranes. One milliliter of undiluted cell suspension was filtered through 5-μm pore size polycarbonate membrane (it4ip, Belgium). The filter was then rinsed with phosphate-buffered saline (PBS) buffer, carefully cut (~0.5 cm × 0.5 cm), and stuck with double face adhesive tape on one side of the microscope Petri dish while avoiding dewetting. On the other side, 50 μL of the opposite mating type diluted 100× suspension (*MAT*a) were dropped and let to settle for 5 min before being rinsed in PBS. Cell–cell force spectroscopic experiments were then performed at room temperature (20 °C) in filtered PBS buffer, using a JPK Nanowizard 4 atomic force microscope combined with an inverted optical microscope (Zeiss Axio Observer Z1 equipped with a Hamamatsu camera C10600, Zeiss AG) and connected to a pressure pump unit and a pressure controller through a microfluidic tubing system (Cytosurge AG).

**FluidFM single-cell experiments.** A single yeast *MAT*a cell was picked up from the glass surface of the Petri dish by approaching the FluidFM probe and applying a negative pressure (−80 kPa). The transfer of the cell on the probe was verified by optical microscopy. The obtained yeast probe was then transferred over the porous membrane and precisely positioned over a single target *MAT*α yeast cell using optical microscopy. Adhesion maps were obtained by recording 16-by-16 force–distance curves on areas of 100-by-100 nm$^2$ with an applied force of 1 nN, a constant approach and retraction speed of 1 μm s$^{-1}$, and a contact time of 1 s. For LR experiments, arrays of 16-by-16 force curves were recorded on 100-by-100 nm$^2$ areas at increasing retraction speeds as follows: 0.2, 1, 5, and 10 μm s$^{-1}$. For some experiments, DTT (Sigma-Aldrich) and the histidyl-modifying agent DEPC were injected at a final concentration of 10 mM.

**Data analysis.** Force spectroscopic data were analyzed with the data processing software from JPK Instruments (Berlin, Germany). The unbinding force and rupture length were extracted from the last specific peak in each force vs extension curve. A specific adhesive event is defined as an event where the retraction segment of the force curve shows a variation in the slope (representing the stretching of the molecular complex) before the rupture point. The frequency of those specific adhesion events, recorded in a map on a 100 × 100 nm$^2$ area on the cell, is defined as the adhesion probability. LR was extracted from the linear slope immediately preceding the rupture event on the force vs time curves. Distribution of the parameters of interest were then plotted and further analyzed with Origin.

**Statistics and reproducibility.** The statistical significance of differences among yeast cell pairs adhesion probabilities was assessed using Student's *t* test on the Origin software (2017). *p* values when differences are significant are provided on graphs and in figure captions. The number of independent cell pairs is also provided in the text and, when appropriate, in figures and corresponding captions. Experiments were reproducible in at least three independent yeast cultures.

**Reporting summary**. Further information on research design is available in the Nature Research Reporting Summary linked to this article.

## Data availability

All data are available from the corresponding authors upon reasonable request. Source data are provided with this paper.

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

## Acknowledgements

Work at the Université catholique de Louvain was supported by the European Research Council (ERC) under the European Union's Horizon 2020 research and innovation program (grant agreement no. 693630), the FNRS-WELBIO (grant no. WELBIO-CR-2015A-05), the National Fund for Scientific Research (FNRS), and the Research Department of the Communauté française de Belgique (Concerted Research Action). Work at John Jay College was supported by the US National Institute of General Medical Sciences grant SC3GM111133. We thank David Alsteens for fruitful discussion.

## Author contributions

All authors designed the experiments and analyzed the data. M.M.-G., F.V., J.D., P.N.L., and Y.F.D. wrote the article. M.M.-G., F.V., and J.D. collected the data.

## Competing interests

The authors declare no competing interests.
