## [Peer Review File · Communications Biology]

Reviewers' comments:

Reviewer #1 (Remarks to the Author):

For the first time of my career as reviewer i am proposing "publish as it is".

The manuscript is clearly written and the figures are illustrative.
The logical thread behind the design of the experiments is evident.
The experimental data are discussed deeply but the same time frankly.
The materials&methods section is exhaustive.

Reviewer #2 (Remarks to the Author):

In this study, Mathelié-Guinlet et al. applied a method based on fluidic force microscopy to investigate cell-to-cell adhesion during yeast mating. They validate their approach with force measurements that confirm the requirement of pheromone stimulation for expression of agglutinins, which mediate the cell-cell adhesion, and later by chemical treatments disrupting known essential regions of each agglutinins, and resulting in impaired cell attachment. The authors then applied the approach and demonstrate the measurement of single agglutinin pairs interactions, the increasing interactions upon extended contact time, as well as the increased adhesive interactions upon mechanical stress. This work provides a convincing demonstration of the potential of the presented approach to investigate interactions between single living cells, down to single pairs of molecules, and to shed new light on a complex biological process.

Overall, the study was well designed and rigorously executed. The manuscript is well written, with both the biological and technical aspects well explained, and the conclusions accurately reflect the experimental data.

I have only a few minor concerns listed below:

1)Although its general meaning is clear, the term «adhesion probability» which is used to describe most results should be shortly defined once, e.g., in the Methods; was there a cut-off value applied to define «non-adhesive events»?

2)The proof-of-concept assessing the induction of agglutinin expression by the pheromones is totally conclusive, and no additional data are necessary. However, if the data were available, it would be interesting to see how the 20- vs the 2-fold increase in agglutinin expression would reflect in the adhesion measurements (MATa α -factor -MATa pair)...

3)In the part investigating prolonged contact time, the authors consistently described the short and extended contacts as «before and after the 30 min contact between cells». Do they mean «with 1s and 30 min contact time between the cells»? Or «always using 1s contact time, but recorded before or after pressing the two cells together for 30 min»? This should be described in more details in the Methods.

4)Page 3, line 62: "Aploid cells"

Page 6, line 116: "binding binding"

Page 13, line 283: "Cytorsurge AG"

Answers to reviewers for the manuscript COMMSBIO-20-2214-T

Reviewer #1 (Remarks to the Author):

For the first time of my career as reviewer i am proposing "publish as it is". The manuscript is clearly written and the figures are illustrative. The logical thread behind the design of the experiments is evident. The experimental data are discussed deeply but the same time frankly. The materials&methods section is exhaustive.

We warmly thank the reviewer for the very positive feedback and acceptance of our manuscript "as it is" for publication in Communications Biology.

Reviewer #2 (Remarks to the Author):

In this study, Mathelié-Guinlet et al. applied a method based on fluidic force microscopy to investigate cell-to-cell adhesion during yeast mating. They validate their approach with force measurements that confirm the requirement of pheromone stimulation for expression of agglutinins, which mediate the cell-cell adhesion, and later by chemical treatments disrupting known essential regions of each agglutinins, and resulting in impaired cell attachment. The authors then applied the approach and demonstrate the measurement of single agglutinin pairs interactions, the increasing interactions upon extended contact time, as well as the increased adhesive interactions upon mechanical stress. This work provides a convincing demonstration of the potential of the presented approach to investigate interactions between single living cells, down to single pairs of molecules, and to shed new light on a complex biological process. Overall, the study was well designed and rigorously executed. The manuscript is well written, with both the biological and technical aspects well explained, and the conclusions accurately reflect the experimental data.

We thank the reviewer for the very positive feedback and have answered point by point the comments below.

I have only a few minor concerns listed below:

1) Although its general meaning is clear, the term «adhesion probability» which is used to describe most results should be shortly defined once, e.g., in the Methods; was there a cut-off value applied to define «non-adhesive events»?

An interaction event is defined as a specific adhesive event when the retraction segment of the force curve shows a variation in the slope (representing the stretching of the molecular complex) before the rupture point. Consequently, only all those specific adhesion events, recorded in a map on a 100 nm x 100 nm area on the cell, are taking into account to calculate the final adhesion probability on the map, i.e. on the cell. This information has been included in the revised manuscript lines 111-112 and in the Methods section, lines 317-321.

2) The proof-of-concept assessing the induction of agglutinin expression by the pheromones is totally conclusive, and no additional data are necessary. However, if the data were available, it would be interesting to see how the 20- vs the 2-fold increase in agglutinin expression would reflect in the adhesion measurements (MAT α α -factor -MAT α pair)...

As pointed by the reviewer, our data were striking and clear enough to show the importance of pheromone induction in both mating type yeasts, leading us to consider any extra controls as not fundamentally necessary. In addition, the invert experiments where MAT α cells will be treated with α -factor and MAT α cells will remain untreated should lead to an increase of 10-20 fold of α -agglutinins expression that might not be statistically distinguishable, under our fluidFM experimental conditions, as compared to a 2-fold increase in α -agglutinins expression in the control presented in Fig.1.

3) In the part investigating prolonged contact time, the authors consistently described the short and extended contacts as «before and after the 30 min contact between cells». Do they mean «with 1s and 30 min contact time between the cells»? Or «always using 1s contact time, but recorded before or after pressing the two cells together for 30 min»? This should be described in more details in the Methods.

We apologize if this point was not crystal clear in the manuscript. One has to clearly distinguish between :

(1) the “probing time” is the additional time, during the force spectroscopy measurements, you let the tip in contact with the sample before retracting it. In a classical approach-retract cycle (AFM force curve), inherent to the system is the time that the tip will take to approach towards the sample, to touch it and then to retract. The “probing time” that an AFM user can adjust is this time during which you touch the sample before the retraction regime

(2) the “prolonged contact time” between cell-pairs is the time of contact between the cells before (or after, for comparison) actually performing the force spectroscopy measurements, i.e. the time that we artificially force the cells to be in contact. It is totally independent of the time needed to record force curves.

This has been clarified in the revised manuscript, in lines 146-147 and 168-169.

4) Page 3, line 62: "Aploid cells"

Page 6, line 116: "binding binding"

Page 13, line 283: "Cytorsurge AG"

These typos have been corrected in the revised manuscript.